# Projecting the future incidence and burden of dengue in Southeast Asia

**Felipe J. Colón-González** [1,2,3,4,5] ✉, **Rory Gibb**[1,2], **Kamran Khan**[6,7], **Alexander Watts**[7,8], **Rachel Lowe** [1,2,3,9,10,11] & **Oliver J. Brady** [1,2,11]

The recent global expansion of dengue has been facilitated by changes in urbanisation, mobility, and climate. In this work, we project future changes in dengue incidence and case burden to 2099 under the latest climate change scenarios. We fit a statistical model to province-level monthly dengue case counts from eight countries across Southeast Asia, one of the worst affected regions. We project that dengue incidence will peak this century before declining to lower levels with large variations between and within countries. Our findings reveal that northern Thailand and Cambodia will show the biggest decreases and equatorial areas will show the biggest increases. The impact of climate change will be counterbalanced by income growth, with population growth having the biggest influence on increasing burden. These findings can be used for formulating mitigation and adaptation interventions to reduce the immediate growing impact of dengue virus in the region.

Dengue is one of the most important mosquito-borne emerging global threats. Since 1995 the number of countries reporting dengue has trebled and an estimated 100–400 million infections now occur annually across more than 120 countries[1]. The burden in affected countries is also increasing with countries in the Americas, Southeast Asia, and the Western Pacific reporting the majority of global cases[2]. The recent spread has been attributed to a variety of factors, including increased urbanisation, increased human movement, and an increasingly favourable climate[1,3]. Public health policy-makers at international, national and local levels need robust estimates of how dengue risk might change in the long-term to plan effective control and mitigation strategies.

Climate constrains the outer limits of the spatial range, as well as the timing and magnitude of the dengue transmission season through its effects on mosquitoes and virus dynamics[4]. Multiple physiological traits of mosquitoes (particularly adult female longevity, fecundity and biting rate) and the virus (extrinsic incubation period) are modulated by temperature with higher temperatures leading to higher transmission, although extremely high temperatures adversely affect survival of adult mosquitoes[5,6]. Precipitation modulates the creation of breeding sites[7], though droughts may also lead to the creation of breeding sites by increasing water storage behaviours[8]. Several previous studies have fitted models to dengue data based solely on temperature and precipitation, then used these to suggest that projected future increases in these variables will lead to geographic expansion, longer transmission seasons, and more intense transmission[5,9–11].

Global urbanisation trends and increasing human mobility are often suggested as drivers of dengue expansion[12,13], but rarely quantitatively analysed. Urbanisation and increases in human population density create new habitat for dengue's primary mosquito vector, *Aedes aegypti*, which preferentially lays eggs in artificial water holding containers in close proximity to humans that form the primary food source for adult females[1]. Human movement at local to regional levels facilitates dengue spread and can re-seed areas where seasonal

[1]Department of Infectious Disease Epidemiology, Faculty of Epidemiology and Population Health, London School of Hygiene & Tropical Medicine, London WC1E 7HT, UK. [2]Centre for Mathematical Modelling of Infectious Diseases, London School of Hygiene & Tropical Medicine, London WC1E 7HT, UK. [3]Centre on Climate Change and Planetary Health, London School of Hygiene & Tropical Medicine, London WC1E 7HT, UK. [4]Tyndall Centre for Climate Change Research, School of Environmental Sciences, University of East Anglia, Norwich NR4 7TJ, UK. [5]Data for Science and Health, Wellcome Trust, London NW1 2BE, UK. [6]Department of Medicine, Division of Infectious Diseases, University of Toronto, Toronto, ON M5S 3H2, Canada. [7]BlueDot, Toronto, ON M5J 1A7, Canada. [8]Esri Canada, Toronto, ON M3C 3R8, Canada. [9]Barcelona Supercomputing Center (BSC), Barcelona 08034, Spain. [10]Catalan Institution for Research and Advanced Studies (ICREA), Barcelona 08010, Spain. [11]These authors contributed equally: Rachel Lowe, Oliver J. Brady. ✉e-mail: Felipe.Colon@lshtm.ac.uk

dynamics cause local extinction[14,15]. At an international level, human movement can spread different dengue serotypes that can increase the risk of re-infection and severe disease[16]. Few studies have considered the effects of future changes in urbanisation on dengue[17,18], and none have projected how future changes in human movement could affect transmission.

One further factor to consider is socioeconomic development. Higher income levels and educational standards, usually measured through the proxy of socioeconomic status, could reduce dengue risk through improved infrastructure to reduce breeding sites, more intensive mosquito control practices, and better uptake and adherence to mosquito preventative practices[19,20]. The example of long-term suppression of dengue infection incidence in Singapore relative to its regional neighbours is frequently cited as the possible role economic development could play in limiting dengue expansion[21].

The majority of previous dengue projection studies have used dengue presence/absence data or mechanistic models parameterised using theoretical knowledge or laboratory experiments[10,17]. Such approaches are useful for defining the geographic and seasonal limits of transmission but often poorly predict dynamics in endemic countries[22] where incidence is constrained by factors such as immunity and demographics[23]. This has led some studies to suggest that changes in future dengue burden will be driven primarily by population growth in endemic areas rather than environmentally driven changes in risk[17,22].

Here, we aim to predict the effect of multiple global change phenomena on future dengue incidence and burden over the period 2020–2099 across Southeast Asia, one of the worst affected regions globally. These results can be used by policy-makers to set appropriate goals for dengue mitigation and identify where investment in new control tools may be needed to build resilience to dengue in the region[24]. Our projections account for changes in climate, population density, human mobility, and gross domestic product (a proxy for socioeconomic development) to reflect the effects of factors known to be relevant for dengue dynamics, and for which data were available for the historical and future periods. The main contributions of these study are twofold. First, our results derive from a model formulated using long-term dengue cases reported at a fine spatial scale across eight countries in Southeast Asia. This provides more robust results than proxy measures of transmission, such as vectorial capacity. Second, our study projects changes in dengue incidence based on different future scenarios of the most important determinants of dengue risk (i.e. climate, urbanisation, socioeconomic development, and human mobility) all of which have been important in shaping the expansion of dengue in previous years.

## Results

We specified a Generalised Additive Mixed Model with a conditional negative binomial distribution for counts of dengue cases reported at the province level across eight countries in Southeast Asia over the period January 2000–December 2017. Six determinants of dengue risk (air temperature, number of consecutive dry days, human population density, human mobility, air travel volume, and GDP) were included in the model as explanatory variables. The number of consecutive dry days (CDD) per month was used as a proxy for water availability for the creation of breeding sites. Within-country human mobility was approximated using a radiation model.

A blocked cross-validation algorithm was used to investigate the predictive ability of the model using data for the reference (historical) period 2000–2017 ($n = 216$ months). The mean absolute error (MAE) was used as a measure of predictive ability because it is a natural and unambiguous measure of average skill magnitude[25]. The selected model had a median cross-validated MAE of 37.5 cases per month across all locations and time steps. This value should be interpreted relative to a total of 5,284,064 dengue cases across all locations over

the study period, and a monthly mean of 114 (range 0–11,212) dengue cases across the region. We note that the MAE was larger in regions and months of the year with a higher number of dengue cases, and was typically lower than the mean number of monthly dengue cases (see Supplementary Information). Analysing the cross-validated MAE as a proportion of the observed cases, the median ratio remained below one for 89% of the years in the series. Mean cross-validated in-sample and out-of-sample Spearman's rank correlation were estimated at 0.82 and 0.78, respectively, with some between-block variation that was larger in the out-of-sample predictions (Supplementary Information).

We present projections of dengue burden and incidence for the period 2020–2099 ($n = 960$ months) based on the most recent Shared Socioeconomic Pathways (SSPs)[26] developed for the Coupled Model Intercomparison Project Phase 6 (CMIP6)[27]. The period 2020-2099 was selected to allow for predictions on complete decadal periods. An ensemble of all available general circulation models, GCM (GFDL-ESM4, IPSL-CM6A-LR, MPI-ESM1-2-HR, MRI-ESM2-0, and UKESM1-0-LL) bias-corrected for the third simulation round of the Inter-Sectoral Impact Model Intercomparison Project[28] was selected to explore multi-model uncertainty. Briefly, GCMs are numerical models representing physical processes to depict the climate using a three dimensional (ocean, cryosphere, and land surface) grid over the world. We considered three different SSPs namely, SSP126, SSP370, and SSP585 to represent a wide range of socioeconomic trends and radiative forcings (Table 1). SSPs are named according to their broad socioeconomic trends and their end-of-century radiative forcing relative to pre-industrial conditions (Table 1). A total of 15 GCM-SSP combinations were used for the projections. We compare our projections to the reference period 2000–2017. We note that the distribution of projected variables is close to the distribution of the observed covariates used for model fitting (Supplementary Information).

### Estimated determinants of dengue risk

We simulated 1000 samples from the posterior distribution of dengue cases to allow for the estimation of uncertainty in our predictions. Consistent with previous research[4,5,20,29–31], we find that increases in temperature, CDD, population density, and international travel increase dengue risk up to a particular value, beyond which risk decreases or plateaus (Fig. 1). We also find a strong protective effects of several variables likely to change in the future, such as increasing GDP and, to a lesser extent, greater within-country human mobility[19]. We also find that areas with very high population densities (>1500 people/km²) are associated with a decrease in dengue risk.

### Long-term regional trends

While the projected future of dengue differs by SSP scenario, there is a consensus that both dengue burden and incidence will rise in the short to medium term, peak sometime this century then begin to decline, likely below historic levels before the end of the century (Fig. 2). We predict a maximum of 69.5 (52.9–76.1) annual cases per 100,000 people or 580,000 (441,000–635,000) annual cases by 2080 under scenario SSP370. Early peaks of up to 59.8 (54.3–63.2) annual cases per 100,000 people or 399,000 (361,000–415,000) annual cases are projected mid century, followed by declines below current-day incidence and burden sometime between 2050 and 2075. The most substantial declines are estimated in the two most extreme SSPs (SSP126 and SSP585) likely for different primary reasons. In SSP126, predicted reductions to 48.2 (46.5–50.0) annual cases per 100,000 or 240,000 (231,000–249,000) annual cases by 2099 are due to more moderate increases in global-mean temperature (reaching 1.7 °C above pre-industrial levels by the end of the century), but more substantial increases in global economic growth[5,32]. In scenario SSP585, predicted reductions to a minimum of 27.6 (8.3–39.4) annual cases per 100,000 people or about 135,000 (41,000–193,000) annual cases are due to the

**Table 1 | Summary of SSP narratives and percentage change in the predictors relative to the historical period**

| Scenario | Socioeconomic assumptions | Temperature | Number of dry days | GDP | Population | Population density | Within-country mobility | International travel |
|---|---|---|---|---|---|---|---|---|
| SSP126 | Low challenges to mitigation and adaptation. Focus on human well-being rather than on economic growth. Reduced income inequality between and within states. Consumption with minimised resource and energy usage. Fast urbanisation in all regions of the world associated with high income growth. Low population growth rate with global population reaching about 7 billion people by 2100 | 2050: 3.9% (0.6 to 8.3) 2080: 4.2% (-0.1 to 10.4) | 2050: 22.5% (-4.8 to 49.1) 2080: 22.4% (-8.7 to 56.4) | 2050: 83.9% (55.6 to 100.3) 2080: 133.8% (100.3 to 155.9) | 2050: 23.1% (17.9 to 25.3) 2080: 5.9% (-7.3 to 17.4) | 2050: 36.1% (33.1 to 37.5) 2080: 23.2% (8.3 to 34.7) | 2050: 63.6% (62.0 to 64.6) 2080: 54.1% (45.3 to 61.4) | 2050: 377.9% (266.1 to 481.6) 2080: 561.7% (488.2 to 613.5) |
| SSP370 | High challenges to mitigation and adaptation. Revival of nationalism and regional conflict. Policies increasingly focused on national and regional security. Decreasing investment in education and technological development. Rising between and within state inequality. Urbanisation constrained by slow economic growth with limited mobility across regions. Poor urban planning makes cities unattractive. High population growth rates with global population reaching 12.6 billion people by 2100 | 2050: 5.9% (0.9 to 13.2) 2080: 10.6% (4.1 to 20.7) | 2050: 27.1% (-11.2 to 71.5) 2080: 30.1% (-2.7 to 69.7) | 2050: 42.8% (34.3 to 47.5) 2080: 56.5% (47.5 to 59.5) | 2050: 44.1% (35.7 to 50.4) 2080: 54.8% (50.7 to 58.9) | 2050: 39.7% (32.4 to 45.7) 2080: 49.8% (46.1 to 53.4) | 2050: 114.3% (99.6 to 127.9) 2080: 143.3% (132.8 to 153.3) | 2050: 419.6% (279.4 to 569.9) 2080: 774.9% (581.1 to 996.2) |
| SSP585 | High challenges to mitigation and low challenges to adaptation. Strong innovation and technological progress. Strong world economic growth and local environmental problems tackled successfully. Social and economic development based on an intensified exploitation of fossil fuels. Fast urbanisation in all regions of the world due to rapid economic growth and technological change. Low population growth rate with global population reaching about 7 billion people by 2100 | 2050: 7.0% (1.0 to 16.6) 2080: 13.4% (5.3 to 26.8) | 2050: 24.3% (-11.1 to 61.0) 2080: 28.3% (-6.4 to 61.9) | 2050: 124.7% (68.5 to 161.6) 2080: 248.1% (161.6 to 308.6) | 2050: 21.9% (16.5 to 24.2) 2080: 4.3% (-8.9 to 15.9) | 2050: 40.6% (36.0 to 42.7) 2080: 28.6% (12.9 to 40.4) | 2050: 62.4% (61.2 to 63.5) 2080: 52.2% (43.3 to 59.7) | 2050: 399.6% (276.4 to 515.0) 2080: 604.4% (522.4 to 661.8) |

Values in brackets indicate the between-year range.

increasing protective effect of strong economic growth, but also considerably hotter global-mean temperatures (reaching 4.9 °C above preindustrial levels) that will become increasingly unfavourable for mosquito survival in some areas[5,32].

### Geographic heterogeneity

While, in the long term, dengue incidence is expected to decline across the region, this trend masks substantial geographic heterogeneity in the direction and magnitude of expected change. Relative to the reference period 2000–2017 (Fig. 3a), the multi-GCM multi-scenario ensemble mean of the predictions suggests increases in dengue incidence across large areas of central Vietnam, Laos, Malaysia, Singapore, Indonesia and the southern half of the Philippines (Fig. 3b) both by 2050 (the average conditions for 2040–2069) and 2080 (the average conditions for 2070–2099). Conversely, we project decreases in incidence across large areas of Cambodia, Thailand, southern Vietnam and the northern half of the Philippines over the same periods. Most models agree with the direction of the mean predicted change (Fig. 3c), across most of the region except for parts of the Philippines and central Vietnam, particularly at later time points.

These changes will result in a shift in how the burden of dengue is distributed between its constituent countries Fig. 4a). Vietnam is projected to shift from having the second highest burden in the region (23% of all cases) in 2000–2017 to having the fourth (15% of all cases) by 2080 as a growing proportion of the burden will occur in Malaysia (21%) and the Philippines (20%). Projected declines in burden in Thailand (from 12% of the burden in 2000–2017 to 5% by 2080) and proportionally stable trends in other countries are predicted to lead to an increasingly polarised distribution of burden across the region with 93% of cases in Indonesia, Philippines, Malaysia and Vietnam and only 7% percent in all other countries combined by 2080. These changes are underscored by the emerging importance of high (but not very high) population density areas (301–1500 people/km², Fig. 4b) where dengue burden will rise from 35 to 49% of all cases. This trend is driven by (1) projected decreases in incidence in very high density cities due to concentration of economic development that is expected to improve control capabilities and infrastructure to reduce mosquito breeding habitat[20]; and (2) increases in the proportion of people living in high but not very high density areas such as city suburbs as rural to urban migration accelerates (Supplementary Information).

### Contribution of different factors to future changes in incidence and burden

Climate, GDP, within country human mobility and population are all projected to undergo substantial change over the next 80 years, but at different levels and with different consequences for dengue risk (Fig. 5). To try to understand the relative contribution of each of these global change phenomena, we conduct a sensitivity analysis of our model predictions where each single variable was held at historical monthly mean values, but all other variables were subject to their projected future changes. This analysis gives insights into how dengue risk would change if future changes in individual dengue risk factors could be prevented or mitigated against.

Our results reveal that future projected changes in population, GDP and climate are likely to play the largest role in future dengue burden (Fig. 5). If climate were to stay at historical levels, but all other factors change as projected, approximately 45,000 fewer cases a year would occur by 2080 compared to our baseline scenario where all variables change as projected. Conversely, if the projected increases in GDP do not occur, approximately 42,000 more cases may occur per year, highlighting the near equal contribution of sustaining economic growth and combating climate change to limiting dengue burden in the region. However, the most influential factor for future growth in dengue burden will be population growth which ensures that even if

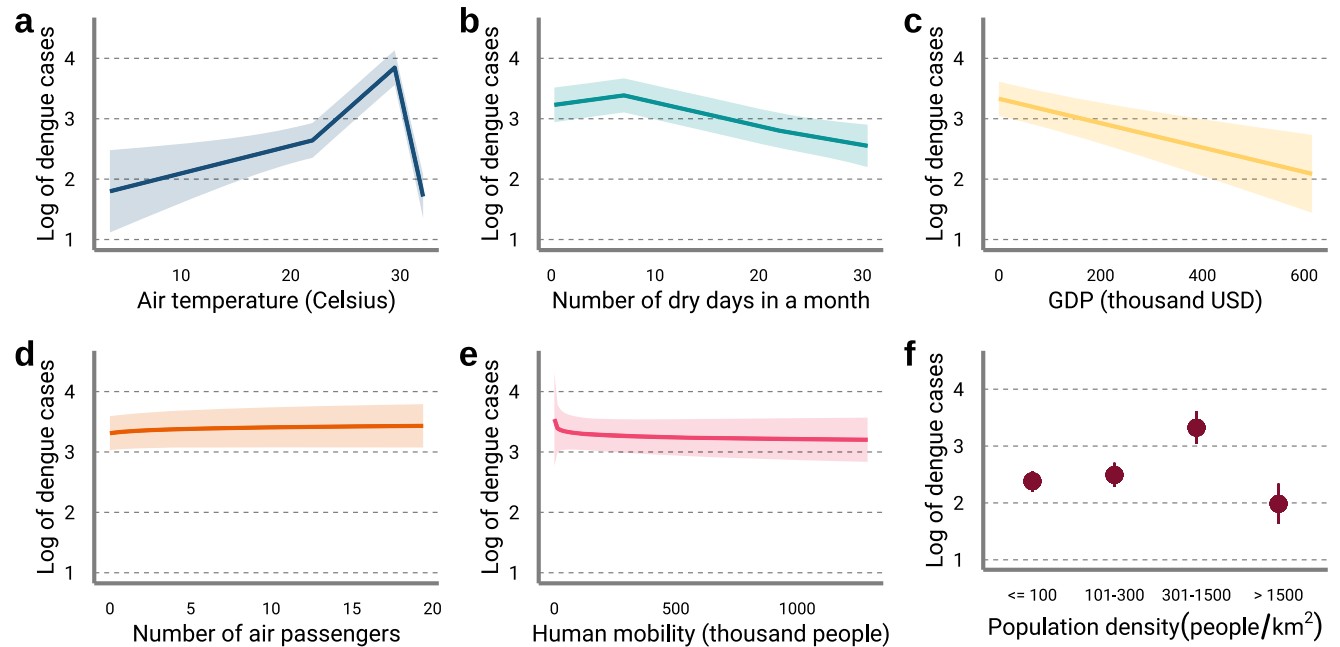

**Fig. 1 | Partial dependency plots of the final model covariates.** Partial dependency plots of **a** air temperature, **b** continuous dry days, **c** gross domestic product, **d** number of air passengers, **e** within-country human mobility, and **f** population density as estimated by the final model. Solid lines and points indicate the mean partial effect. Shaded areas and error bars indicate the 95% confidence interval of the partial effects. Source data are provided as a Source data file.

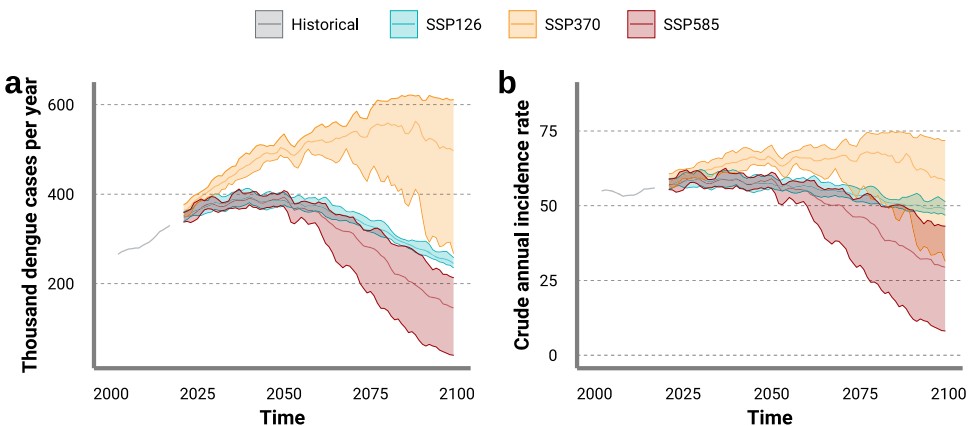

**Fig. 2 | Influence of shared socioeconomic pathway (SSP) scenario on the long-term regional trends of future dengue burden and future dengue incidence in Southeast Asia.** Multi-model ensemble mean of the predicted long-term trends in **a** annual dengue cases and **b** crude annual dengue incidence per 100,000 people averaged across Southeast Asia by SSP scenario (*n* = five general circulation models examined over the period January 2020 to December 2099). Solid curves indicate the multi-model ensemble mean. Shaded areas indicate the spread of the general circulation model-specific posterior means. Source data are provided as a Source data file.

dengue incidence rates remain stable or decrease, the number of cases may still be expected to rise (contrasting Fig. 2a and Fig. 2b). If the number of people living in South East Asia remained at historic levels around 58,000 fewer cases would occur per year compared to a scenario where population increases as expected. This importance of population growth would further increase if, as projected, increases in population also result in unfavourable changes in population density (Fig. 5).

When we disaggregated these results by country, we observed similar patterns in Indonesia, Laos, Malaysia, and the Philippines particularly by mid century with substantial between-country heterogeneity (Supplementary Information). A protective effect of climate was observed in Cambodia, Thailand, and Vietnam, particularly towards the end of the century.

## Discussion

Accounting for the combined effects of climatic, demographic and socio-economic factors is crucial for a better understanding of the potential effects of climate change on future dengue risk. We used one of the best possible data sets of epidemiological, climatic, and non-climatic data currently available to systematically derive projections of dengue burden and dengue incidence to provide valuable information for long-term decision-making and planning. Our findings indicate that both dengue burden and dengue incidence will peak sometime this century in Southeast Asia, before declining to historical levels or below depending on the prevailing climatic and socioeconomic conditions. These effects will vary considerably between countries and across varying levels of population density, in agreement with previous research[5,9,17,18].

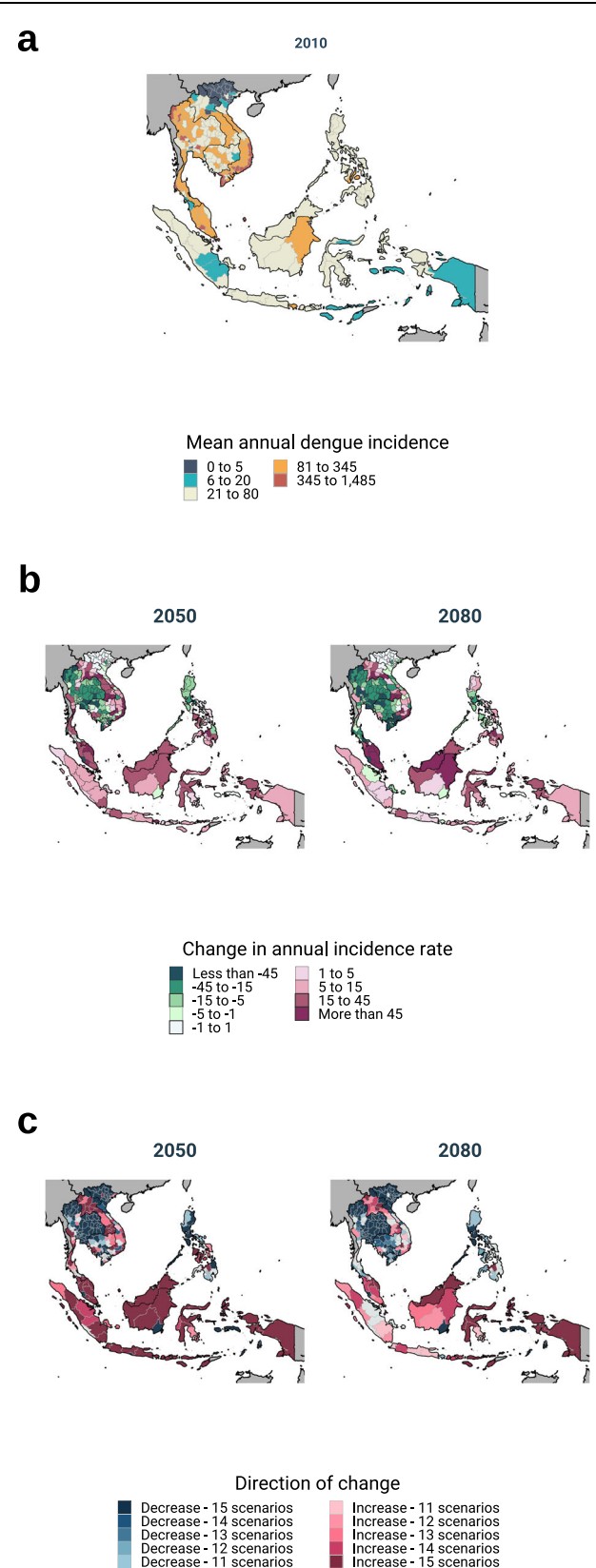

**Fig. 3 | Spatial heterogeneity patterns of the observed data and the model predictions. a** Mean annual crude dengue incidence rate per 100,000 over the period 2000–2017 (equal intervals in the logarithmic scale; $n = 216$ months). **b** Predicted change in the mean annual crude dengue incidence rate over the periods 2050 and 2080 ($n =$ five general circulation models examined over 360 months respectively). Green colours indicate a decrease and purple colours indicate an increase in incidence. **c** Between-model agreement in the direction of the predicted change in future dengue incidence. Blue colours indicate a decrease in incidence and purple colours indicate an increase. Source data are provided as a Source data file.

as a route cause of this difference through changes use of air conditioning and sealed housing and differential human movement behaviours that reduce exposure to mosquito bites. Some larger-scale ecological observational studies from dengue-endemic areas have also identified associations between increased dengue risk and lower socioeconomic characteristics that are not explained by other environmental factors[19,33–35]. Combined, these studies suggest a closer focus on indicators more closely related to dengue risk (e.g. provision of reliable piped water and refuse collection) than general proxies of economic development and future dengue projection work should focus on how to translate generic projections of GDP into specific infrastructure and housing changes to improve our understanding of how future increases in dengue risk can be mitigated. The need to understand links between aggregate economic measures, such as GDP, willingness to pay and implementation of dengue-specific interventions will only become more important as new vaccines and novel vector control tools are now beginning to be used at scale[24,36–38].

The finding that very high density cities are projected to confer the lowest risk was also unexpected and has considerable implications for future projected risk in highly urbanised areas such as South East Asia. Over the past few decades as mobility has increased at the rural-urban fringe, the differences in dengue exposure between rural and urban areas has decreased[15,39]. It is also difficult to distinguish whether dengue is associated with the process of urbanisation (i.e. land use change and the disruption it brings) or urban land per se. The association between dengue and construction sites might explain why we project relatively mature very high density city cores to confer lower dengue risk than more actively changing high density areas outside the city core[40]. More work is needed on analysis of long term (10+ years) high resolution dengue combined with data on infrastructure and land use to understand how dengue risk changes during the rural-to-urban transition.

This is one of the first studies to quantitatively compare how future dengue risk will change in response to major global changes in climate, economic development, urbanisation, and human mobility. Our empirical modelling approach allowed us to directly project future changes in dengue incidence and burden as opposed to indirect measures such as vectorial capacity, climate suitability, epidemic potential, or reproduction number[5,10,17]. Our findings provide a more detailed understanding of how mitigation actions against future rises in dengue risk could be planned. In particular, limiting global-mean temperature below 2 °C above preindustrial levels, greater focus on control and urban planning in high density suburban areas combined with strong economic growth may result in important public health benefits for the region. While large increases in global-mean temperature might also reduce dengue transmission in the region, this should not be interpreted as a benefit since non-optimal temperatures could lead to other pervasive effects of climate change such as the invasion of warmer-adapted disease vectors[41], heat-related excess mortality[42], and loss of labour productivity[43].

Our projections of dengue risk should not be interpreted as forecasts of what will happen in the future but as scenarios of what might happen based on a set of pre-specified assumptions. In all of

Research examining climate change impacts on future dengue risk often neglects the effects of social, environmental, and economic factors that determine the level of risk in observed past and present dengue case count trends[5,9,11]. Evidence of a link between economic factors and dengue risk has been growing since the initial observation of substantially differing levels of dengue virus exposure either side of the US–Mexico border. In this case, economic factors were attributed

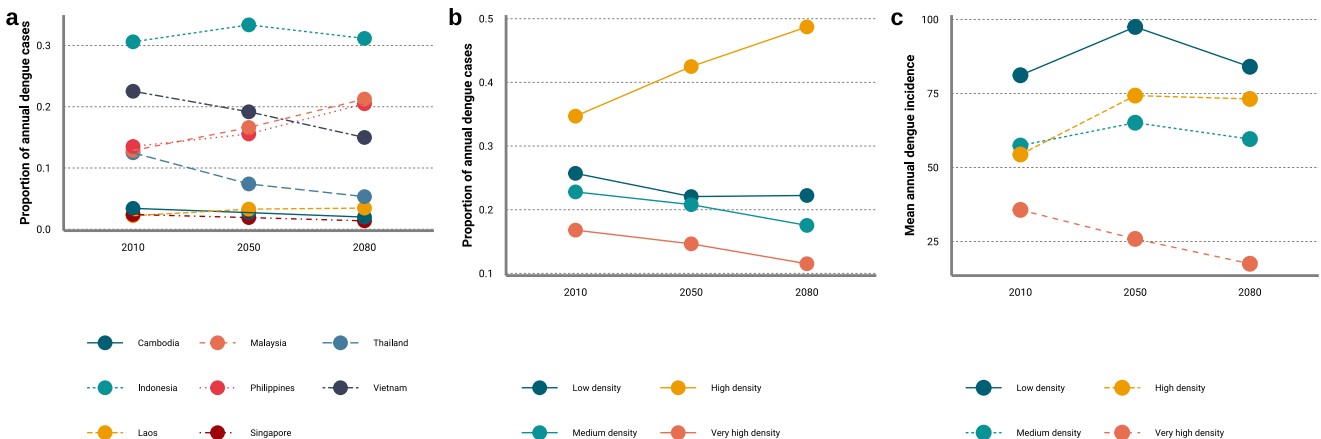

**Fig. 4 | Shifts in the distribution of the burden and incidence of dengue across Southeast Asia.** Shifts in how the mean annual number of dengue cases is projected to be distributed across **a** constituent countries and **b** population density levels (across the whole region) by time period. **c** Shifts in how dengue the mean annual dengue incidence incidence rate per 100,000 people for Southeast Asia is projected to be distributed across different population density levels by time period (*n* = 5 general circulation models examined over 216 months for the period 2010, and over 360 months for the periods 2050 and 2080). Source data are provided as a Source data file.

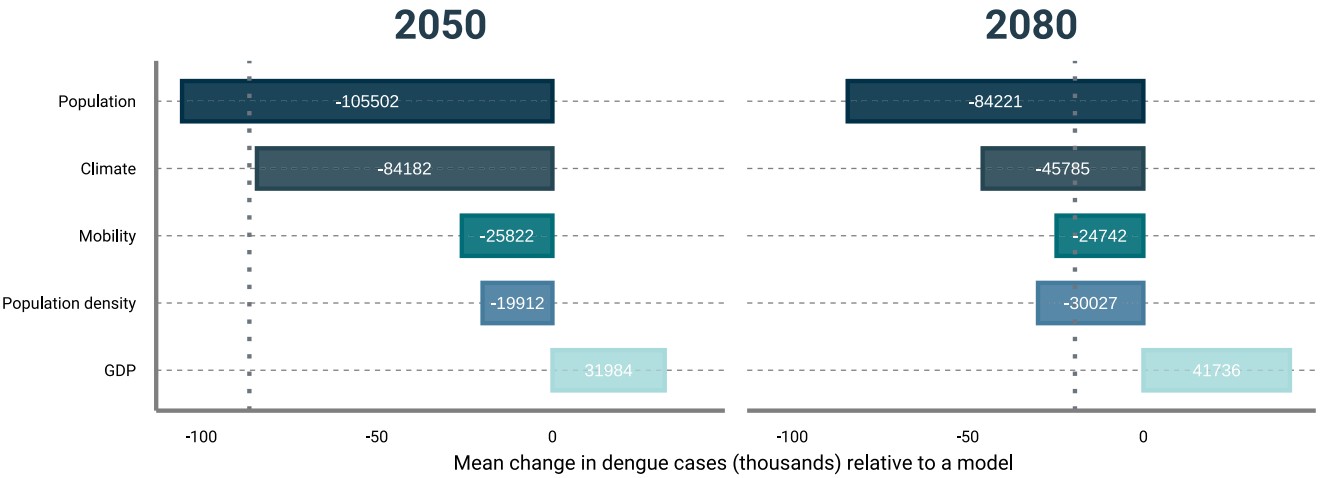

**Fig. 5 | Sensitivity analysis of the final model predictions.** Projected changes in mean annual dengue cases (thousands) relative to a model where all variables change in the future, under the assumption that one group of predictors (population, climate, human mobility, population density, and gross domestic product (GDP)) remains constant at their historical monthly mean values. The vertical dashed lines indicate the mean annual number of dengue cases relative to a model where all variables change in the future (*n* = 5 general circulation models examined over 360 months). Source data are provided as a Source data file.

these scenarios, we assume interventions remain at current levels to isolate the effects of various global change phenomena on dengue risk. This approach avoids making strict assumptions about the effectiveness and scale of implementation of emerging dengue vaccines and vector control tools that are yet to be proven effective at operational scales (e.g., *Wolbachia* population replacement programmes[24]), and indicates our results can be used for separate modelling exercises that focus on the question of quantifying what level of adoption of novel tools are required to mitigate future increases in dengue transmission.

There are several limitations of this study. The dengue data used here does not account for changes in surveillance, and so improvements over time could accentuate increases in reported cases. This situation may lead to biases in our estimations of dengue incidence in the future. Population immunity has also been shown to play a strong role in limiting the upper bounds of dengue incidence and shaping regional and seasonal dynamics but was not explicitly included in our model[23]. The clinical presentation of dengue is age-dependent, and the age distribution of the population will continue to change over the

next century. Previous research simulating age-specific case counts demonstrated that age shifts in reported severe dengue cases in Thailand between 1981 and 2017 could be attributed to the shifting age demography of the population[44]. Whilst it would be important to consider the age structure of the population in our study, currently available epidemiological data from cohort does not provide information to undertake an age-stratification of cases of clinically diagnosed dengue in the region.

Our study does not account for the effects of COVID-19 restrictions on dengue risk which appear to have reduced dengue risk in the region, albeit for an unknown duration[45]. With national and international travel rebounding fast in the region, any protective effect of human mobility restrictions might be limited to a short period and might be undone due to higher population susceptibility leading to above average incidence over the next few years. Our statistical model does not incorporate potential interactions between variables or complex non-linearities which might result in different levels of future dengue risk being estimated. More advanced modelling approaches

could be implemented in future studies to represent these relationships more effectively. Our projections of dengue risk are determined by our selection of GCMs and climate change scenarios. While there is considerable between-GCM agreement in our estimates, we predict large between-scenario differences. One of the major challenges for decision makers acting on these projections will be to understand how to act in the face of these differing scenarios which are reflective of real-world uncertainties in how climate policy and socioeconomic development might drive future environmental change. Despite these limitations, our study provides actionable information for policy-makers and public health professionals to mitigate the immediate threats of increasing dengue burden in the region.

## Methods

### Epidemiological dengue data

Monthly dengue cases for the period January 2000 to December 2017 at the province (i.e., administrative level 1) level were obtained from project TYCHO (http://www.tycho.pitt.edu), epidemiological surveillance bulletins, Ministries of Health web pages for Cambodia, Indonesia, Laos, Malaysia, the Philippines, Singapore, Thailand, and Vietnam (Supplementary Table 1). The definition of a dengue case comprised of an unspecified mixture of suspected and laboratory confirmed cases. Epidemiological data for Cambodia and Vietnam for the period 2011–2017 were obtained at the national and annual levels and so they were linearly downscaled at the province and monthly levels by keeping the monthly fractional share of each province's dengue cases relative to the period 2000–2010 constant[46]. Missing observations ($n = 7361$) were omitted from the model fitting restricting the analysis to the set of fully-observed observations. Data for the period January 2018–December 2019 were unavailable at the time of data acquisition.

**Population data.** Global gridded annual population counts on a $0.5 \times 0.5$ degree latitude-longitude grid were obtained from the ISI-MIP data base (https://esg.pik-potsdam.de/projects/isimip/) for historical (2000–2015) and future (2015–2099) periods[47]. Future projections were obtained for three SSPs (i.e. SSP126, SSP370, and SSP585). To align the historical population data with our time period of historical dengue data, population counts for the period 2005–2017 were linearly interpolated using the approx function in R. Because some historical population estimates were significantly different to those published by the United Nations (UN), we calibrated our country-level population estimates to match UN estimates for the year 2005[48]. Population density (people/km²) was computed for all periods and all SSPs by dividing population counts for each administrative unit by their surface area using the raster R package[49]. Population density estimates were used to define administrative units as low density (i.e. ≤100 people/km²), moderate density (i.e. 101–300 people/km²), high density (i.e., 301–1500 people/km²), and very high density areas (i.e., >1501 people/km²)[18,50].

**Gross Domestic Product (GDP) data.** Historical and future GDP estimates were obtained from a previous modelling study that mapped GDP (based on purchasing power parity in 2005 US dollars) to a resolution of 30 arc-seconds based on a variety of remotely sensed socioeconomic covariates[51]. Historical GDP estimates were only available for the year 2005 and future estimates available at 10-year intervals for the period 2020–2100. Province-specific annual estimates for the period 2000–2017 were calculated by multiplying the estimated GDP for the year 2005 by the relative annual increase in GDP between 2000 and 2017 from the World Bank[52].

**Human mobility data.** We generated province-specific annual estimates of within-country mobility flux using two general movement models, a naive (unparameterised) gravity model[53] and the radiation model[54]. The gravity model assumes that the potential gravity flux ($G_{ij}$)

between two locations $i$ and $j$ is influenced only by the populations of the source and destination locations and the intervening distance, such that

$$G_{ij} = \frac{p_i p_j}{d_{ij}} \tag{1}$$

where $p_i$ is the population of the source location, $p_j$ is the population of the destination, and $d_{ij}$ is the great circle distance between the centroids of the source and destination provinces. The radiation model includes an additional constraint to account for the competing attractiveness of other destinations within the same distance radius[54], such that potential radiation flux ($R_{ij}$) between two locations is predicted by:

$$R_{ij} = T_i \frac{p_i p_j}{(p_i + s_{ij})(p_i + p_j + s_{ij})} \tag{2}$$

where $T_i$ is the number of inhabitants that start their journey from location $i$ (assumed to be all inhabitants, i.e. $T_i = p_i$[54]); and $s_{ij}$ is the total population within the circle of radius $d_{ij}$ centred at location $i$ excluding the source and destination populations.

Within each country we generated annual matrices of pairwise predicted mobility fluxes between all pairs of provinces (2000–2017, and 2018–2099 for each SSP scenario) using gravity and radiation models. These were summarised as annual province-specific mean within-country gravity and radiation flux (i.e. mean predicted flux between the focal province and all other provinces in the same country and year). We restricted mobility calculations to within-country fluxes only due to the difficulty of accounting for the geographically variable impacts of international borders on human movement.

**Air passenger volume data.** Air passenger volume data for flights within and between countries were obtained from the International Air Transport Association (IATA) at the province level and at monthly time steps for January 2010–December 2017. Data were obtained for provinces with an airport. Data for the period 2000–2009 were calculated multiplying the number of passengers for the year 2010 by the country-specific increase in passengers relative to 2010 as estimated by the World Bank[55]. Future air passenger volumes for the period 2020–2099 were calculated by multiplying national air passenger volumes in 2017 by the annual Asia-Pacific region-wide mean annual growth rate as estimated by IATA[56]. Future air passenger volumes were assumed to correspond to the middle of the road SSP2. SSP-specific air passenger volumes were calculated multiplying the expected volume for SSP2 by the difference in population between SSP2 and each of the other SSPs.

The annual estimates of air passenger volumes were calculated for individual airports (i.e. specific point locations) to produce province-level annual estimates of air passenger volumes. We geographically disaggregated annual passengers from each airport to the surrounding provinces. We used a gravity model approach, that assumes that the proportion of passengers travelling from any airport a to destination province $j$ is proportional to the population of $j$ divided by the intervening distance (i.e. more nearby and more populated centres receive the greatest numbers of passengers). This was calculated and summed across all airports within each country, such that:

$$F_j = \sum_{\alpha} \omega_{j,a} A_a \tag{3}$$

where $F_j$ is the total number of passengers arriving at location $j$ in a given year from across all airports, $A_a$ is the total passengers at airport $a$, and $\omega_{j,a}$ is a vector of gravity weights that describe the proportion of passengers that travel from airport a to location $j$. The weights were

calculated as:

$$\omega_{j,a} = \frac{\left(\frac{p_j}{d_{aj}}\right)}{\sum \left(\frac{p_j}{d_{aj}}\right)} \qquad (4)$$

where $p_j$ is the population of destination $j$ and $d_{aj}$ is the population-weighted mean distance between airport $a$ and all 1-km$^2$ grid cells within destination province $j$, which is constant across the time series and calculated using WorldPop population rasters for 2015[57]. As with the mobility models, we assumed no movement of passengers across international borders.

### Historical climate data

Historical climate data were obtained from the Copernicus Climate Data Store (https://cds.climate.copernicus.eu/). Hourly near-surface air temperature (K), hourly rainfall flux (kg/m$^2$/s), near surface specific humidity (dimensionless), and near surface wind speed (m/s) were retrieved from the bias-corrected near-surface meteorological variables (WFDE5) derived from the fifth generation of the European Centre for Medium-Range Weather Forecasts atmospheric reanalyses (ERA5) at a 0.5 × 0.5 degree latitude-longitude grid for the period January 1999 to December 2017. Data were aggregated at monthly steps using the Climate Data Operators (CDO) software[58]. Spatial aggregation at the province level was conducted in R version 4.1 using the `raster` R package[49]. The number of tropical nights and the number of summer days in a month were calculated using the `eca_tr` and `eca_csu` functions in CDO[58]. The number of wet and dry days in a month were calculated using the `eca_cwd` and `eca_cdd` functions in CDO[58].

**Future climate data.** Bias-corrected global daily mean surface temperature (K), and total precipitation (kg/m$^2$/s) data were retrieved from the ISI-MIP data base (https://esg.pik-potsdam.de/projects/isimip/) on a 0.5 × 0.5 degree latitude-longitude grid for the only five general circulation models (GFDL-ESM4, IPSL-CM6A-LR, MPI-ESM1-2-HR, MRI-ESM2-0, UKESM1-0-LL) available across three Tier-1 CMIP6 scenarios[59] (arranged by their end of the century radiative forcing level: SSP126, SSP370, SSP585). Data were retrieved for the period 2020-2099 at monthly time steps. Data for the period 2018–2019 were not retrieved to include complete decades. The processed epidemiological, climatic, socioeconomic, and demographic data are available at the Centre for Open Science (https://osf.io/85xwq/).

**Statistical model.** The number of dengue cases $Y_{i,t}$ for administrative unit $i = 1, \cdots, I = 246$ at month $t = 1, \cdots, T = 216$ was modelled using a generalised additive mixed model (GAMM) with a negative binomial distribution. The general algebraic definition of the model[9,18] is given by:

$$\log(\mu_{i,t}) = \alpha + \log(P_{i,a[t]}) + \sum_{k=1}^{K} f(X_{i,t,k}) + \sum_{l=1}^{L-1} \beta_l U_{i,a[t]} + \gamma G_{i,d[t]}$$
$$+ \delta \log(M_{i,a[t]}) + \epsilon \log(T_{i,a[t]}) + \zeta_{i,a[t]} + \eta_{m[t]} + \nu_i + u_i, \qquad (5)$$

where $\alpha$ denotes the intercept; $\log(P_{i,a[t]})$ indicates the logarithm of the population in administrative unit $i$ at year $a[t]$, included as an offset; $X$ is a matrix of $k = 1, \cdots, K = 3$ piecewise functions of climate variables (air temperature, and number of dry days) defined as linear regression splines $f$; $U$ is a categorical variable of different population density levels ($l$) for administrative unit $i$ at year $a[t]$ with regression coefficients $\beta$; $G$ denotes the GDP for each administrative unit and decade $d[t]$ with regression coefficient $\gamma$; $\log(M_{i,a[t]})$ indicates the natural logarithm of human mobility with coefficient $\delta$; and $\log(T_{i,a[t]})$

indicates the natural logarithm of air passenger volume with coefficient $\epsilon$. Constrains to the linear regression splines were based on exploratory analyses using cubic regression splines. We preferred linear splines over cubic splines because the latter are built from basis functions defined from the training data and our projections of climatic and non-climatic variables extend beyond that range making cubic splines inappropriate for prediction. Delayed effects of climate were accounted for by incorporating a three-month moving average centred at a lag of one month. Annual anomalies were accounted for using unstructured random effects for each year ($\zeta_i, a[t]$). Seasonal trends are accounted for using cubic regression splines $\eta$ for each calendar month $m[t]$[60]. Unknown confounding factors and spatial dependencies were incorporated using spatially structured ($\nu_i$) and unstructured ($u_i$) random effects for each administrative unit $i$. Spatially structured random effects were specified using a Gaussian Markov Random Field smooth, which represents the spatial dependence structure of areas that share a boundary[60]. Models were fitted in R version 4.1 using the `mgcv` package[60].

### Model evaluation

Blocked cross-validation[61] was used to evaluate the predictive ability of the model. We divided the data set into $K = 216$ training and testing sets. Each test set comprised of blocks of $n = 24$ contiguous observations. The natural order of the observations was kept within each block. The predictive ability of the model was evaluated using the mean absolute error (MAE) for each block. In-sample and out-of-sample Spearman's rank correlation coefficients were also calculated for each block. In a sensitivity analysis, we fitted models using the six predictors in isolation, as well as all their possible combinations ($n = 64$ models) using a blocked cross-validation algorithm (see 'Methods')[9,62].

## Data availability

Data sets generated and/or analysed during the current study are available within the paper, the provided link or are appended as Supplementary Data. Source data are provided with this paper. The processed epidemiological, climatic, socioeconomic and demographic data are available at the Centre for Open Science OSF data repository (https://osf.io/85xwq/, https://doi.org/10.17605/OSF.IO/85XWQ).

## Code availability

The code for the statistical model is available for download from GitHub https://github.com/FelipeJColon/Projecting_dengue_SEA or from the Centre for Open Science OSF data repository https://osf.io/85xwq/, https://doi.org/10.17605/OSF.IO/85XWQ.

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

## Acknowledgements

F.J.C.-G., R.G., R.L. and O.J.B. were supported by the UK Space Agency Dengue Forecasting Model Satellite-based System (D-MOSS). R.L. was supported by a Royal Society Dorothy Hodgkin Fellowship.

## Author contributions

All authors contributed to the study design, strategy components, interpretation and manuscript writing. F.J.C.-G., R.L. and O.J.B. conceived and developed an initial design for the study. F.J.C.-G., R.G., and A.W. verified and post-processed the data. F.J.C.-G. wrote the code with notable contributions from R.G. F.J.C.-G., R.G., A.W., R.L. and O.J.B. analysed model results. F.J.C.-G., R.G., R.L. and O.J.B. designed the visualisation of results. F.J.C.-G. drafted the article. All authors revised and commented on the article and approved the final version.

## Competing interests

The authors declare no competing interests.
