## [Peer Review File · Nature Communications]

REVIEWER COMMENTS

Reviewer #1 (Remarks to the Author):

In this study, the authors combined climate and non-climate factors associated with dengue incidence in order to project dengue burden in Southeast Asia for the remainder of the 21st Century. Fitting a monthly model to eight countries at the province-level showed that dengue incidence will increase steadily over the next several decades before declining by the end of the century. Overall, this is a strong study with interesting implications. The use of socioeconomic predictors to project dengue burden is particularly novel and interesting. Specific comments are below.

1. The introduction section previous studies on this topic and some of their limitations. In the last paragraph of this section, the authors should state more explicitly what sets this study apart. For example, is the greatest motivating strength the model form, the data sources, the two together, or something else?
2. The first two paragraphs of the results section may be clearer to the readers if switched. The section states that projections were made based on scenarios from socioeconomic and climate forecasts and then follows by describing the statistical model developed to predict dengue. Switching the two would be easier to follow sequentially, stating that the statistical model was fit to predict dengue and then applied to scenarios based on socioeconomic and climate forecasts.
3. The beginning of the results section could use some additional explanation of the GCMs for readers not as familiar with this topic. The strings of letters and numbers are not easy to understand out of context. An additional sentence or two qualitatively describing what they are and how they fit into this study would help clarify.
4. In the section "estimated determinants of dengue risk," interpreting the RMSE would help with additional context. Was 263 found averaging across all locations and months, or averaged across months, aggregated for all locations? It would also be beneficial to see how this compares to any summary statistics of monthly cases. I realize that RMSE does not perfectly translate to absolute error, but in this context would be helpful in interpreting whether this is high or low accuracy.
5. The methods section describes the dengue data, and after seeing the supplemental table, it seems that not all 12*18 months were available for all eight counties. Were missing observations imputed or omitted from model fitting? It would also be clearer to the readers to state in the main text how many months of data were or were not available.
6. When describing the statistical model, check the coefficients listed with each predictor. The mobility and air passenger coefficients appear to be mislabeled in the text.
7. The model selection section lists seven potential predictors considered for inclusion, six of which are included in the final model. The results section states that 252 models were considered; clarify how 252 model forms were derived. Including/excluding the seven parameters only yields $2^7 = 128$ models.
8. The model selection also lists three "other variables" that were considered for inclusion after the previous seven variables were applied in model fitting. Why were these variables considered after initial model fitting rather than alongside the other variables during the initial model fitting?

Reviewer #2 (Remarks to the Author):

Colón-González et al. project the future incidence and burden of dengue in SE Asia under

a variety of scenarios. The manuscript is well-written. I do not have some comments regarding the design of the study.

1) The authors use 2000-2017 as the reference period and 2020-2099 for prediction. Please justify such split of the two periods. In particular, why the period of 2018-2019 is excluded.

2) Relatedly, as the prediction covers the period of 2019-2021, it would be good the authors could demonstrate the potential influence of Covid-19 pandemic on factors such as human mobility and thus the prediction of dengue incidence and burden. It is possible that the impact of Covid-19 pandemic on the prediction over longer periods (i.e. by 2050 and 2080) may be marginal; but should not be neglected. I would suggest the authors to use data in 2000-2019 (if data available) for reference and 2022-2099 for prediction.

3) Contribution of different factors to future changes is the key to targeting dominant drivers. The authors made the analysis for SE Asia as shown in Figure 5, how does the findings vary at finer scale(s)? For example, the dominant drivers may be different across eight countries by 2050 and 2080. It would be interesting to show the heterogeneities of the dominant drivers across countries and two periods.

Reviewer #3 (Remarks to the Author):

The paper by Colón-González fits a statistical model to province-level monthly dengue case counts 8 Southeast Asian countries and then uses this model to project the dengue burden up to 2099 considering projected changes in a set of socioeconomic (mainly GDP), demographic, and climate variables. I'm generally skeptical of the usefulness of such long terms projections, not only because of the large sources of uncertainty but because it is quite likely that vaccines and vector-control interventions will become available before that time horizon. However, I think the question is broadly relevant (understanding how dengue burden will change as a function of multiple changing drivers) and shorter-term projections are useful. I also appreciate that, acknowledging the uncertainty of projections in the covariates used, the authors consider several GCM-SSP combinations. Nevertheless, I have several concerns about the manuscript and think it is not publishable in its current form. I summarize my main concerns below.

My main concern is that, other than stating the RMSE of the best fitting model, the paper does not present results on model fit or predictive performance. This makes it very difficult to evaluate the validity and robustness of the results presented since overfitting is always a big risk in this type of model.

At the very least I'd like to see plots of the correlation between predicted and observed data points, for both the in-sample and out-of-sample observations. Furthermore, given that the principal output of this study is projections over periods that extend well beyond the observation time (216 months of data and 960 months of projection), I think that leave-one-out cross-validation is not enough. Blocked cross-validation (over blocks of space and time) might give a better sense of how competing models might perform when extrapolating.

Similarly, I'd like to see more results to better understand what drives the projections. How much do the associations between covariates and outcome vary by country and over time? To what extent does the distribution of projected covariates match the distribution of covariates used for model fitting (in other words, how much is being extrapolated beyond the observed range of covariates?).

I also have some concerns about how to interpret estimates of dengue burden that are aggregated (over all age groups) and not age-stratified. The clinical presentation of dengue is known to be age-dependent, and the age distribution of the population will continue to change over the next century. This, at the very least, should be discussed.

Along the same line, the model does not consider changes in the age structure of the population, even though they are key determinants of both transmission dynamics and disease burden. Did the authors consider including birth rate, life expectancy, or other metrics of population structure in their models?

Response to Reviewers comments

Manuscript reference number: NCOMMS-22-23253

Title: Projecting the future incidence and burden of dengue in Southeast Asia

We wish to thank you for your comments as they have led to improvements to the manuscript. All changes are described below and can be summarised as follows:

- Added a paragraph indicating the added value of our study
- Incorporated a description of the general circulation models for non-specialists
- Included an analysis of the predictive ability of the model stratified by administrative unit, month of the year, and year in the time series.
- Corrected and clarified the number of alternative models explored in the analysis
- Explained our rationale for restricting predictions to the period 2020-2099
- Added a figure to the supplementary information section showing the country-stratified contribution of the covariates in the model
- Discussed the utility of our scenario-based projections
- Replaced our time series cross-validation algorithm with a block cross-validation algorithm
- Discussed the age dependency of the clinical presentation of dengue cases.

Reviewer #1 (Remarks to the Author):

In this study, the authors combined climate and non-climate factors associated with dengue incidence in order to project dengue burden in Southeast Asia for the remainder of the 21st Century. Fitting a monthly model to eight countries at the province-level showed that dengue incidence will increase steadily over the next several decades before declining by the end of the century. Overall, this is a strong study with interesting implications. The use of socioeconomic predictors to project dengue burden is particularly novel and interesting. Specific comments are below.

Thank you for the positive evaluation of our manuscript.

1. The introduction section previous studies on this topic and some of their limitations. In the last paragraph of this section, the authors should state more explicitly what sets this study apart. For example, is the greatest motivating strength the model form, the data sources, the two together, or something else?

We thank the reviewer for this suggestion. We have included the following text in the main body of the manuscript.

“The main contributions of these study are two-fold. First, our results derive from a model formulated using long-term dengue cases reported at a fine spatial scale across eight countries in Southeast Asia. This provides more robust results than proxy measures of transmission, such as vectorial capacity. Second, this study projects changes in dengue incidence based on different future scenarios of the most important determinants of dengue risk (i.e., climate, urbanisation, socioeconomic development, and human mobility) all of which have been important in shaping the expansion of dengue in previous years.”

2. The first two paragraphs of the results section may be clearer to the readers if switched. The

section states that projections were made based on scenarios from socioeconomic and climate forecasts and then follows by describing the statistical model developed to predict dengue. Switching the two would be easier to follow sequentially, stating that the statistical model was fit to predict dengue and then applied to scenarios based on socioeconomic and climate forecasts.

Thank you for the suggestion. We switched the paragraphs in the Results section as suggested.

3. The beginning of the results section could use some additional explanation of the GCMs for readers not as familiar with this topic. The strings of letters and numbers are not easy to understand out of context. An additional sentence or two qualitatively describing what they are and how they fit into this study would help clarify.

As suggested, we included an additional explanation of the GCMs as follows:

“Briefly, GCMs are numerical models representing physical processes to depict the climate using a three-dimensional (ocean, cryosphere, and land surface) grid over the world.”

4. In the section “estimated determinants of dengue risk,” interpreting the RMSE would help with additional context. Was 263 found averaging across all locations and months, or averaged across months, aggregated for all locations? It would also be beneficial to see how this compares to any summary statistics of monthly cases. I realize that RMSE does not perfectly translate to absolute error, but a this context would be helpful in interpreting whether this is high or low accuracy.

Thank you for highlighting this issue. For this revision, we used the mean absolute error (MAE) instead of the RMSE as the former is a more natural and unambiguous measure of error. We included the following explanation of its interpretation to the main body of the manuscript:

“The mean absolute error (MAE) was used as a measure of predictive ability because it is a natural and unambiguous measure of average skill magnitude. The selected model had median cross-validated MAE of 37.5 cases per month averaged across all locations and time steps. This value should be interpreted relative to a total of 5,284,064 dengue cases across all locations over the study period, and a monthly mean of 114 (range 0-11,212) dengue cases across the region. We note that the MAE was larger in regions and months of the year with a higher number of dengue cases, and was typically lower than the mean number of monthly dengue cases (see Supplementary information). Analysing the MAE as a proportion of the observed cases, the median cross-validated MAE remained below one 89% of the years in the series.”

We also added the following figures to the Supplementary Information section:

Supplementary Figure 1. Spatiotemporal patterns in the observed number of dengue cases and the mean absolute error.

Spatial patterns of a. the mean number of dengue cases and b. median cross-validated mean absolute error (both on a logarithmic scale) per administrative unit across Southeast Asia over the period 2000 to 2017. c. Mean monthly seasonal trends in the mean number of dengue cases (blue) and median cross-validated mean absolute error (red) per country across the historic period 2000 to 2017.

Supplementary Figure 2. Interannual variation in the mean absolute error.

Interannual variation in the median cross-validated mean absolute error across Southeast Asia over the period 2000 to 2017. The grey lines indicate the median cross-validated mean absolute error for each of the alternative models explored in the study. The orange line indicates the median cross-validated mean absolute error of the selected model. The blue line indicates the mean number of observed dengue cases.

5. The methods section describes the dengue data, and after seeing the supplemental table, it seems that not all 12*18 months were available for all eight counties. Were missing observations imputed or omitted from model fitting? It would also be clearer to the readers to state in the main text how many months of data were or were not available.

We added a clarification in the text to indicate that missing observations were omitted from the model fitting.

“Missing observations (n = 7,361) were omitted from the model fitting restricting the analysis to the set of fully-observed observations.”

6. When describing the statistical model, check the coefficients listed with each predictor. The mobility and air passenger coefficients appear to be mislabelled in the text.

Thank you for bringing this to our attention. We have checked and amended the coefficients.

7. The model selection section lists seven potential predictors considered for inclusion, six of which are included in the final model. The results section states that 252 models were considered; clarify how 252 model forms were derived. Including/excluding the seven parameters only yields $2^7 = 128$ models.

We have removed the *Model selection* section and replaced it with a *Model evaluation* section that we believe is more appropriate for the current version of the manuscript. In this new section, we have corrected the calculations for the number of fitted models.

“In a sensitivity analysis, we fitted models using the six predictors in isolation, as well as all their possible combinations (n = 64 models) using a blocked cross-validation algorithm.”

8. The model selection also lists three “other variables” that were considered for inclusion after the previous seven variables were applied in model fitting. Why were these variables considered after initial model fitting rather than alongside the other variables during the initial model fitting?

When fitting the initial predictive model, we wanted to include the major predictors known to be relevant for dengue dynamics (air temperature, number of consecutive dry days [a proxy for water availability for breeding sites], human population density, human mobility, air travel volume, and GDP). We tested whether other factors such as wind speed and specific humidity improved the predictive ability of the model in a series of preliminary analyses but none of them helped improving the model skill. To avoid confusion, we have decided not to mention this preliminary analysis in this revised version of the manuscript.

Reviewer #2 (Remarks to the Author):

Colon-Gonzalez et al. project the future incidence and burden of dengue in SE Asia under a variety of scenarios. The manuscript is well-written. I do not have some comments regarding the design of the study.

We thank the reviewer for their positive comments.

1) The authors use 2000-2017 as the reference period and 2020-2099 for prediction. Please justify such split of the two periods. In particular, why the period of 2018-2019 is excluded.

Dengue data for the period 2018-2019 were not freely available for any of the countries at the time of the data acquisition and so they were not included in the historical period. The period 2020-2099 was selected to allow for predictions on complete decadal periods. This is now mentioned in the manuscript:

“The period 2020-2099 was selected to allow for predictions on complete decadal periods.”

2) Relatedly, as the prediction covers the period of 2019-2021, it would be good the authors could demonstrate the potential influence of Covid-19 pandemic on factors such as human mobility and thus the prediction of dengue incidence and burden. It is possible that the impact of Covid-19 pandemic on the prediction over longer periods (i.e., by 2050 and 2080) may be marginal; but should not be neglected. I would suggest the authors to use data in 2000-2019 (if data available) for reference and 2022-2099 for prediction.

As stated above, at the time of the data acquisition, dengue records for the period 2018 onwards were not freely available for any of the countries. We acknowledge that this is a limitation of our study as the effects of COVID-19 restrictions on dengue risk appear to have reduced dengue risk in the region, albeit for an unknown duration

(<https://www.sciencedirect.com/science/article/pii/S1473309922000251>). We note however that national and international travel is rebounding fast in the region

(<https://www.iata.org/en/pressroom/2022-releases/2022-06-09-01/>) and so any protective effects of human mobility restrictions might be limited to 1-2 years and might be cancelled out due to higher population susceptibility leading to above average incidence over the next few years.

3) Contribution of different factors to future changes is the key to targeting dominant drivers. The authors made the analysis for SE Asia as shown in Figure 5, how does the findings vary at finer scale(s)? For example, the dominant drivers may be different across eight countries by 2050 and 2080. It would be interesting to show the heterogeneities of the dominant drivers across countries and two periods.

We have disaggregated the results presented in Figure 5 by country (below) and presented them in the Supplementary Information.

Supplementary Figure 6. Sensitivity analysis of the model predictions stratified by country.

Country-stratified projected changes in mean annual dengue cases (thousands) relative to a model where all variables change in the future, under the assumption that one group of predictors (climate, GDP, human mobility, and population) remains constant at their historical monthly mean values.

Reviewer #3 (Remarks to the Author):

1) The paper by Colón-González fits a statistical model to province-level monthly dengue case counts 8 Southeast Asian countries and then uses this model to project the dengue burden up to 2099 considering projected changes in a set of socioeconomic (mainly GDP), demographic, and climate variables. I'm generally skeptical of the usefulness of such long term projections, not only because of the large sources of uncertainty but because it is quite likely that vaccines and vector-control interventions will become available before that time horizon. However, I think the question is broadly relevant (understanding how dengue burden will change as a function of multiple changing drivers) and shorter-term projections are useful. I also appreciate that, acknowledging the uncertainty of projections in the covariates used, the authors consider several GCM-SSP combinations. Nevertheless, I have several concerns about the manuscript and think it is not publishable in its current form. I summarize my main concerns below.

We agree with the reviewer that these projections should not be interpreted as forecasts of what will happen with dengue in the future. Rather their utility is conditional on the set of assumptions ingrained in each future projection scenario, i.e., if emissions are not curbed by x%, then we would expect a y% increase in dengue transmission. In all of these scenarios, we assume interventions remain at their current levels to isolate the effects of various global change phenomena on dengue risk. This approach i) avoids making strict assumptions about the effectiveness and scale of implementation of emerging dengue vaccines and vector control tools that are yet to be proven at operational scales and ii) means our results can be used for separate modelling exercises that focus on the question of quantifying what level of adoption of novel tools are required to mitigate future increases in dengue transmission. We have clarified the utility of these scenario-based projections in a new section in the discussion as follows:

“Our projections of dengue risk should not be interpreted as forecasts of what will happen in the future but as scenarios of what might happen based on a set of pre-specified assumptions. In all of these scenarios, we assume interventions remain at current levels to

isolate the effects of various global change phenomena on dengue risk. This approach avoids making strict assumptions about the effectiveness and scale of implementation of emerging dengue vaccines and vector control tools that are yet to be proven effective at operational scales (e.g., Wolbachia population replacement programmes²⁴), and indicates our results can be used for separate modelling exercises that focus on the question of quantifying what level of adoption of novel tools are required to mitigate future increases in dengue transmission.”

2) My main concern is that, other than stating the RMSE of the best fitting model, the paper does not present results on model fit or predictive performance. This makes it very difficult to evaluate the validity and robustness of the results presented since overfitting is always a big risk in this type of model.

At the very least I'd like to see plots of the correlation between predicted and observed data points, for both the in-sample and out-of-sample observations. Furthermore, given that the principal output of this study is projections over periods that extend well beyond the observation time (216 months of data and 960 months of projection), I think that leave-one-out cross-validation is not enough. Blocked cross-validation (over blocks of space and time) might give a better sense of how competing models might perform when extrapolating.

We thank you for bringing this to our attention. We have replaced our Time Series Cross-Validation algorithm with a Blocked Cross-Validation algorithm to ensure the robustness of our estimates. We have also calculated the Bayesian Information Criteria (BIC) as well as in-sample and out-of-sample Spearman's rank correlation coefficients of our predictions. This is now explained in the text as follows:

“Blocked cross-validation⁶⁰ was used to evaluate the predictive ability of the model. We divided the data set into $K=216$ training and testing sets. Each test set comprised of blocks of $n=24$ contiguous observations. The natural order of the observations was kept within each block. The predictive ability of the model was evaluated using the mean absolute error (MAE) for each block. In-sample and out-of-sample Spearman's rank correlation coefficients were also calculated for each block.”

3) Similarly, I'd like to see more results to better understand what drives the projections. How much do the associations between covariates and outcome vary by country and over time? To what extent does the distribution of projected covariates match the distribution of covariates used for model fitting (in other words, how much is being extrapolated beyond the observed range of covariates?).

We have now disaggregated the results presented in Figure 5 by country and presented them in the Supplementary Information to represent how the effects of each of the predictors is expected to vary by country and time. We have also included a series of histograms per covariate in the Supplementary Information section to show the distribution of the covariates for both the historical and future periods. As can be observed from the histograms, the distribution of projected covariates is roughly within the range of the historical period.

Supplementary Figure 6. Sensitivity analysis of the model predictions stratified by country.

Country-stratified projected changes in mean annual dengue cases (thousands) relative to a model where all variables change in the future, under the assumption that one group of predictors (climate, GDP, human mobility, and population) remains constant at their historical monthly mean values.

Supplementary Figure 4. Distribution of historical and projected data.

Histograms of the historical (navy blue) and projected (light blue) data for each of the six predictive variables explored in the study.

4) I also have some concerns about how to interpret estimates of dengue burden that are aggregated (over all age groups) and not age-stratified. The clinical presentation of dengue is known to be age-dependent, and the age distribution of the population will continue to change over the next century. This, at the very least, should be discussed. Along the same line, the model does not consider changes in the age structure of the population, even though they are key determinants of both transmission dynamics and disease burden. Did the authors consider including birth rate, life expectancy, or other metrics of population structure in their models?

We agree that shifting age demographics are likely to be very important for changing severity of dengue cases in Southeast Asia in the future. However, our measure of burden does not consider disease severity, only the number of reported cases. If we were modelling severe cases / DHF /

severe dengue, then we would agree that the age distribution would be important to consider in fitting and projection due to age-dependent disease severity effects.

To the best of our knowledge, current data from cohort and modelling studies does not support any substantial age-dependent effect of being reported as a case of clinically diagnosed dengue (any level of severity) given infection. This was supported by a recent national study in Thailand that was able to disentangle the effects of age-dependent severity and age-dependent reporting (Figure 3 in <https://doi.org/10.1073/pnas.2115790119>). We have, added the following in the discussion about changing risk of severe dengue due to demographics:

“The clinical presentation of dengue is age-dependent, and the age distribution of the population will continue to change over the next century. Previous research simulating age-specific case counts demonstrated that age shifts in reported severe dengue cases in Thailand between 1981 and 2017 could be attributed to the shifting age demography of the population.⁴⁵ Whilst it would be important to consider the age structure of the population in our study, currently available epidemiological data from cohort does not provide information to undertake an age-stratification of cases of clinically diagnosed dengue in the region.”

REVIEWERS' COMMENTS

Reviewer #1 (Remarks to the Author):

The revised manuscript has addressed all of my previous concerns. I think it is suitable for publication.

Reviewer #2 (Remarks to the Author):

The authors have fully addressed my concerns and the manuscript have been well revised.